# Distributed Balanced Clustering via Mapping Coresets

**MohammadHossein Bateni**
Google NYC
bateni@google.com

**Aditya Bhaskara**
Google NYC
bhaskaraaditya@google.com

**Silvio Lattanzi**
Google NYC
silviol@google.com

**Vahab Mirrokni**
Google NYC
mirrokni@google.com

## Abstract

Large-scale clustering of data points in metric spaces is an important problem in mining big data sets. For many applications, we face explicit or implicit size constraints for each cluster which leads to the problem of clustering under capacity constraints or the "balanced clustering" problem. Although the balanced clustering problem has been widely studied, developing a theoretically sound distributed algorithm remains an open problem. In this paper we develop a new framework based on "mapping coresets" to tackle this issue. Our technique results in first distributed approximation algorithms for balanced clustering problems for a wide range of clustering objective functions such as $k$-center, $k$-median, and $k$-means.

## 1 Introduction

Large-scale clustering of data points in metric spaces is an important problem in mining big data sets. Many variants of such clustering problems have been studied spanning, for instance, a wide range of $\ell_p$ objective functions including the $k$-means, $k$-median, and $k$-center problems. Motivated by a variety of big data applications, distributed clustering has attracted significant attention over the literature [11, 4, 5]. In many of these applications, an explicit or implicit size constraint is imposed for each cluster; e.g., if we cluster the points such that each cluster fits on one machine, the size constraint is enforced by the storage constraint on each machine. We refer to this as *balanced* clustering. In the setting of network location problems, these are referred to as *capacitated* clustering problems [6, 16, 17, 10, 3]. The distributed balanced clustering problem is also well-studied and several distributed algorithms have been developed for it in the context of large-scale graph partitioning [21, 20][1]. Despite this extensive literature, none of the distributed algorithms developed for the balanced version of the problem have theoretical approximation guarantees. The present work presents the first such distributed algorithms for a wide range of balanced clustering problems with provable approximation guarantees. To acheive this goal, we develop a new technique based on *mapping coresets*.

A coreset for a set of points in a metric space is a subset of these points with the property that an approximate solution to the whole point-set can be obtained given the coreset alone. An augmented concept for coresets is the notion of *composable coresets* which have the following property: for a collection of sets, the approximate solution to the union of the sets in the collection can be obtained given the union of the composable coresets for the point sets in the collection. This notion was

| MapReduce model | | |
| --- | --- | --- |
| Problem | Approximation | Rounds |
| $L$-balanced $k$-center | O(1) | O(1) |
| $k$-clustering in $\ell_p$ | $O(p)$ | O(1) |
| $L$-balanced $k$-clustering in $\ell_p$ | (O(p),2) | O(1) |
| Streaming model | | |
| Problem | Approximation | Passes |
| $L$-balanced $k$-center | O(1) | O(1) |
| $k$-clustering in $\ell_p$ | $O(p)$ | O(1) |
| $L$-balanced $k$-clustering in $\ell_p$ | (O(p),2) | O(1) |

Table 1: Our contributions, all results hold for $k < n^{1/2-\epsilon}$, for constant $\epsilon > 0$. We notice that for the $L$-balanced $k$-clustering $(p)$ general we get a bicriteria optimization (we can potentially open $2k$ centers in our solutions).

formally defined in a recent paper by Indyk et al [14]. In this paper, we augment the notion of composable coresets further, and introduce the concept of *mapping coresets*. A mapping coreset is a coreset with an additional mapping of points in the original space to points in the coreset. As we will see, this will help us solve balanced clustering problems for a wide range of objective functions and a variety of massive data processing applications, including streaming algorithms and MapReduce computations. Roughly speaking, this is how a mapping coreset is used to develop a distributed algorithm for the balanced clustering problems: we first partition the data set into several blocks in a specific manner. We then compute a coreset for each block. In addition, we compute a mapping of points in the original space to points in the coreset. Finally, we collect all these coresets, and then solve the clustering problem for the union of the coresets. We can them use the (inverse) map to get back a clustering for the original points.

**Our Contributions.** In this paper, we introduce a framework for solving distributed clustering problems. Using the concept of mapping coresets as described above, our framework applies to *balanced* clustering problems, which are much harder than their unrestricted counterparts in terms of approximation.

The rough template of our results is the following: given a single machine $\alpha$-approximation algorithm for a clustering problem (with or without balance constraints), we give a distributed algorithm for the problem that has an $O(\alpha)$ approximation guarantee. Our results also imply streaming algorithms for such clustering problems, using sublinear memory and constant number of passes. More precisely, we consider balanced clustering problems with an $\ell_p$ objective. For specific choice of $p$, it captures the commonly used $k$-center, $k$-median and $k$-means objectives. Our results are also very robust—for instance, bicriteria approximations (violating either the number of clusters or the cluster sizes) on a single machine can be used to give distributed bicriteria approximation algorithms, with a constant loss in the cost. This is particularly important for balanced versions of $k$-median and $k$-means, for which we know of constant factor approximation to the cost only if we allow violating one of the constraints. (Moreover, mild violation might not be terribly bad in certain applications, as long as we obtain small cost.)

Finally, other than presenting the first distributed approximations for balanced clustering, our general framework also implies constant-factor distributed approximations for a general class of uncapacitated clustering problems (for which we are not aware of distributed algorithms with formal guarantees). We summarize our new results in Table 1.

**Related Work.** The notion of coresets has been introduced in [2]. In this paper, we use the term coresets to refer to an augmented notion of coresets, referred to as "composable coresets" [14]. The notion of (composable) coresets are also related to the concept of mergeable summaries that have been studied in the literature [1]. The main difference between the two is that aggregating mergeable summaries does *not* increase the approximation error, while in the case of coresets the error amplifies. The idea of using coresets has been applied either explicitly or implicitly in the streaming model [12, 2] and in the MapReduce framework [15, 18, 5, 14]. However, none of the previous work applies these ideas for balanced clustering problems.

There has been a lot of work on designing efficient distributed algorithms for clustering problems in metric spaces. A formal computation model for the MapReduce framework has been introduced by Karloff et al. [15]. The first paper that studied clustering problems in this model is by Ene et al. [11], where the authors prove that one can use an $\alpha$ approximation algorithm for the $k$-center or $k$-median problem to obtain a $4\alpha + 2$ and a $10\alpha + 3$ approximation respectively for the $k$-center or $k$-median problems in the MapReduce model. Subsequently Bahmani et al. [4] showed how to implement $k$-means++ efficiently in the MapReduce model. Finally, very recently, Balcan et al. [5] demonstrate how one can use an $\alpha$ approximation algorithm for the $k$-means or $k$-median problem to obtain coresets in the distributed (and MapReduce) setting. They however do not consider the balanced clustering problems or the general set of clustering problems with the $\ell_p$ objective function.

The literature of clustering in the streaming model is also very rich. The first paper we are aware of is due to Charikar et al. [7], who study the $k$-center problem in the classic streaming setting. Subsequently Guha et al. [12] give the first single pass constant approximation algorithm to the $k$-median problem. Following up on this, the memory requirements and the approximation factors of their result were further improved by Charikar et al. in [8].

Finally, capacitated (or balanced) clustering is well studied in approximation algorithms [6, 16, 9], with constant factors known in some cases and only bicriteria in others. Our results may be interpreted as saying that the capacity constraints may be a barrier to approximation, but are not a barrier to parallelizability. This is the reason our approximation guarantees are bicriteria.

## 2 Preliminaries

In all the problems we study, we will denote by $(V, d)$ the metric space we are working with. We will denote $n = |V|$, the number of *points* in $V$. We will also write $d_{uv}$ as short hand for $d(u, v)$. Given points $u, v$, we assume we have an oracle access to $d_{uv}$ (or can compute it, as in geometric settings). Formally, a clustering $\mathcal{C}$ of a set of points $V$ is a collection of sets $C_1, C_2, \ldots, C_r$ which partition $V$. Each cluster $C_i$ has a center $v_i$, and we define the '$\ell_p$ cost' of this clustering as

$$\mathsf{cost}_p(\mathcal{C}) := \left( \sum_i \sum_{v \in C_i} d(v, v_i)^p \right)^{1/p}. \tag{1}$$

When $p$ is clear from the context, we will simply refer to this quantity as the cost of the clustering and denote it $\mathsf{cost}(\mathcal{C})$.

Let us now define the $L$-balanced $k$-clustering problem with $\ell_p$ cost.

**Definition 1** ($L$-balanced $k$-clustering ($p$))**.** *Given $(V, d)$ and a size bound $L$, find a clustering $\mathcal{C}$ of $V$ which has at most $k$ clusters, at most $L$ points in each cluster, and cluster centers $v_1, \ldots, v_k$ so as to minimize $\mathsf{cost}_p(\mathcal{C})$, the $\ell_p$ cost defined in Eq. (1).*

The case $p = 1$ is the capacitated $k$-median and with $p = \infty$ is also known as the capacitated $k$-center problem (with uniform capacities).

**Definition 2** (Mapping and mapping cost)**.** *Given a multiset $S$ and a set $V$, we call a bijective function $f : V \to S$ a mapping from $V$ to $S$ and we define the cost of a mapping as $\sum_{v \in V} d(v, f(v))^p$.*

**Definition 3** (Clustering and optimal solution)**.** *Given a clustering problem $\mathcal{P}$ with an $\ell_p$ objective, we define $OPT_{\mathcal{P}}$ as the cost of the optimal solution to $\mathcal{P}$.*

## 3 Mapping coreset framework

The main idea behind our distributed framework is a new family of coresets that help in dealing with balanced clustering.

**Definition 4** ($\delta$-mapping coreset)**.** *Given a set of points $V$, a $\delta$-mapping coreset for a clustering problem $\mathcal{P}$ consists of a multiset $S$ with elements from $V$, and a mapping from $V$ to $S$ such that the total cost of the mapping is upper bounded by $\delta \cdot OPT_{\mathcal{P}}^p$. We define the size of a $\delta$-mapping coreset as the number of distinct elements in $S$.*

Note that our definition does not prescribe the size of the mapping coreset – this can be a parameter we choose. We now define the composability of coresets.

**Definition 5** (Composable $\delta$-mapping coreset). *Given disjoint sets of points $V_1, V_2, \ldots, V_m$, and corresponding $\delta$-mapping coresets $S_1, S_2, \ldots, S_m$, the coresets are said to be* composable *if we have that $\cup_i S_i$ is a $2^p \delta$-mapping coreset for $\cup_i V_i$ (the overall map is the union of those for $V_1, \ldots, V_m$).*

**Remark.** The non-trivial aspect of showing that coresets compose comes from the fact that we compare the cost of mapping to the cost of $\text{OPT}_{\mathcal{P}}$ on the *union* of $V_i$ (which we need to show is not too small). Our main theorem is now the following

**Theorem 1.** *Let $V$ be a set of points and suppose $L, k, p \geq 1$ are parameters. Then for any $U \subseteq V$, there exists an algorithm that takes $U$ as input, and produces a $2^p$-mapping coreset for the $L$-balanced $k$-clustering $(p)$ problem for $U$. The size of this coreset is $\tilde{O}(k)$,[2] and the algorithm uses space that is quadratic in $|U|$. Furthermore, for any partition $V_1, V_2, \ldots, V_r$ of $V$, the mapping coresets produced by the algorithm on $V_1, V_2, \ldots, V_r$ compose.*

### 3.1 Clustering via $\delta$-mapping coresets

The theorem implies a simple general framework for distributed clustering:

1. Split the input into $m$ chunks arbitrarily (such that each chunk fits on a machine), and compute a (composable) $2^p$-mapping coreset for each of the chunks. For each point in the coreset, assign a multiplicity equal to the number of points mapped to it (including itself).

2. Gather all the coresets (and multiplicities of their points) into one machine, and compute a $k$-clustering of this multiset.

3. Once clusters for the points (and their copies) are found, we can 'map back', and find a clustering of the original points.

The idea is that in each chunk, the size of the coreset will be small, thus the union of the coresets is small (and hence fits on one machine). The second step requires care: the clustering algorithm should work when the points have associated multiplicities, and use limited memory. This is captured as follows.

**Definition 6** (Space-efficient algorithm). *Given an instance $(V, d)$ for a $k$-clustering problem in which $V$ has $N$ distinct points, each with some multiplicity, a sequential $\alpha$-approximation algorithm is called space-efficient if the space used by the algorithm is $O(N^2 \cdot poly(k))$.*

The framework itself is a very natural one, thus the key portions are the step of finding the mapping coresets that (a) have small mapping cost and (b) compose well on *arbitrary* partition of the input, and that of finding space efficient algorithms. Sections 4 and 5 give details of these two steps. Further, because the framework is general, we can apply many "levels" of it. This is illustrated below in Section 3.2.

To prove the correctness of the framework, we also need to prove that moving from the original points in a chunk to a coreset with multiplicities (as described in (1)) does not affect us too much in the approximation. We prove this using a general theorem:

**Theorem 2.** *Let $f : V \mapsto S$ be a bijection. Let $\mathcal{C}$ be any clustering of $V$, and let $\mathcal{C}'$ denote the clustering of $S$ obtained by applying a bijection $f$ to the clustering $\mathcal{C}$. Then there exists a choice of centers for $\mathcal{C}'$ such that $\text{cost}(\mathcal{C}')^p \leq 2^{2p-1}(\text{cost}(\mathcal{C})^p + \mu_{total})$, where $\mu_{total}$ denotes $\sum_{v \in V} d(v, f(v))^p$.*

In our case, if we consider the set of points in the coreset with multiplicities, the mapping gives a bijection, thus the above theorem applies in showing that the cost of clustering is not much more than the "mapping cost" given by the bijection. The theorem can also be used in the opposite direction, as will be crucial in obtaining an approximation guarantee.

**Preserving balanced property.** The above theorem allows us to move back and forth (algorithmically) between clusterings of $V$ and (the coreset with multiplicities) $S$ as long as there is a small-cost mapping. Furthermore, since $f$ is a bijection, we have the property that if the clustering was balanced in $V$, the corresponding one in $S$ will be balanced as well, and vice versa.

**Putting things together.** Let us now see how to use the theorems to obtain approximation guarantees. Suppose we have a mapping $f$ from $V$ to the union of the coresets of the chunks (called

$S$, which is a multi set), with total mapping cost $\mu_{\text{total}}$. Suppose also that we have an $\alpha$ space-efficient approximation algorithm for clustering $S$. Now we can use the Theorem 2 to show that in $S$, there exists a clustering whose cost, raised to the $p$-th power, is at most $2^{2p-1}(\text{cost}(\mathcal{C})^p + \mu_{\text{total}})$. This means that the approximation algorithm on $S$ gives a clustering of cost (to the $p$th power) $\leq 2^{2p-1}\alpha^p(\text{cost}(\mathcal{C})^p + \mu_{\text{total}})$. Finally, using Theorem 2 in the opposite direction, we can map back the clusters from $S$ to $V$ and get a an upper bound on the clustering cost (to the $p$th power) of $2^{2p-1}(2^{2p-1}\alpha^p(\text{cost}(\mathcal{C})^p + \mu_{\text{total}}) + \mu_{\text{total}})$. But now using Theorem 1, we know that for the $f$ in our algorithm, $\mu_{\text{total}} \leq 2^p \text{cost}(\mathcal{C})^p$. So plugging this into the bound above, and after some manipulations (and taking $p$th roots) we obtain that the cost of the final clustering is $\leq 32\alpha\text{cost}(\mathcal{C})$. The details of this calculation can be found in the supplementary material.

**Remark.** The approximation ratio (i.e., $32\alpha$) seems quite pessimistic. In our experiments, we have observed (if we randomly partition the points initially) that the constants are much better (often at most 1.5). The slack in our analysis arises mainly because of Theorem 2, in which the worst case in the analysis is very unlikely to occur in practice.

## 3.2 Mapping Coresets for Clustering in MapReduce

The above distributed algorithm can be placed in the formal model for MapReduce introduced by Karloff et al. [15].

The model has two main restrictions, one on the total number of machines and another on the memory available on each machine. In particular, given an input of size $N$, and a sufficiently small $\gamma > 0$, in the model there are $N^{1-\gamma}$ machines, each with $N^{1-\gamma}$ memory available for the computation. As a result, the total amount of memory available to the entire system is $O(N^{2-2\gamma})$. In each round a computation is executed on each machine in parallel and then the outputs of the computation are shuffled between the machines.

In this model the efficiency of an algorithm is measured by the number of the 'rounds' of MapReduce in the algorithm. A class of algorithms of particular interest are the ones that run in a constant number of rounds. This class of algorithms are denoted $\mathcal{MRC}^0$.

The high level idea is to use coreset construction and a sequential space-efficient $\alpha$-approximation algorithm (as outlined above). Unfortunately, this approach does not work as such in the MapReduce model because both the coreset construction algorithm, and the space-efficient algorithm, require memory quadratic in the size of their input. Therefore we perform multiple 'levels' of our framework.

Given an instance $(V, d)$, the MapReduce algorithm proceeds as follows:

1. Partition the points arbitrarily into $2n^{(1+\gamma)/2}$ sets.
2. Compute the composable $2^p$-mapping coreset on each of the machines (in parallel) to obtain $f$ and the multisets $S_1, S_2, \ldots, S_{2n^{(1+\gamma)/2}}$, each with roughly $\widetilde{O}(k)$ distinct points.
3. Partition the computed coreset again into $n^{1/4}$ sets.
4. Compute composable $2^p$-mapping coresets on each of the machines (in parallel) to obtain $f'$, and multisets $S_1', S_2', \ldots, S_{n^{1/4}}'$, each with $\widetilde{O}(k)$ distinct points.
5. Merge all the $S_1', S_2', \ldots, S_{n^{1/4}}'$ on a single machine and compute a clustering using the sequential space-efficient $\alpha$-approximation algorithm.
6. Map back the points in $S_1', S_2', \ldots, S_{n^{1/4}}'$ to the points in $S_1, S_2, \ldots, S_{2n^{(1+\gamma)/2}}$ using the function $f'^{-1}$ and obtain a clustering of the points in $S_1, S_2, \ldots, S_{2n^{(1+\gamma)/2}}$.
7. Map back the points in $S_1, S_2, \ldots, S_{2n^{(1+\gamma)/2}}$ to the points in $V$ using the function $f^{-1}$ and thus obtain a clustering of the initial set of points.

Note that if $k < n^{1/4-\epsilon}$, for constant $\epsilon > \gamma$, at every step of the MapReduce, the input size on each machine is bounded by $n^{(1-\gamma)/2}$ and thus we can run our coreset reduction and a space-efficient algorithm (in which we think of the $\text{poly}(k)$ as constant – else we need minor modification). Furthermore if $n^{1/4-\epsilon} \leq k < n^{(1-\epsilon)/2}$, for constant $\epsilon > \gamma$, we can exploit the trade-off between number of rounds and approximation factor to get a similar result (refer to the supplement for details).

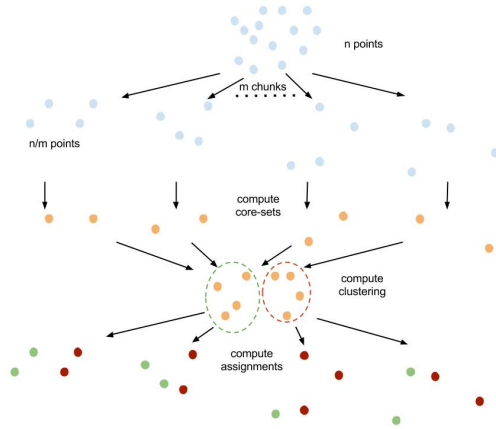

Figure 1: We split the input into $m$ parts, compute mapping coresets for each part, and aggregate them. We then compute a solution to this aggregate and map the clustering back to the input.

We are now ready to state our main theorem in the MapReduce framework:

**Theorem 3.** *Given an instance $(V, d)$ for a $k$-clustering problem, with $|V| = n$ and a sequential space-efficient $\alpha$ approximation algorithm for the (L-balanced) $k$-clustering $(p)$ problem, there exists a MapReduce algorithm that runs in $O(1)$ rounds and obtains an $O(\alpha)$ approximation for the (L-balanced) $k$-clustering $(p)$ problem, for $L, p \geq 1$ and $0 < k < n^{(1-\epsilon)/2}$ (constant $\epsilon > 0$).*

The previous theorem combined with the results of Section 5 gives us the results presented in Table 1. Furthermore it is possible to extend this approach to obtain *streaming algorithms* via the same techniques. We defer the details of this to the supplementary material.

## 4 Coresets and Analysis

We now come to the proof of our main result—Theorem 1. We give an algorithm to construct coresets, and then show that coresets constructed this way compose.

**Constructing composable coresets.**

Suppose we are given a set of points $V$. We first show how to select a set of points $S$ that are *close* to each vertex in $V$, and use this set as a coreset with a good mapping $f$. The selection of $S$ uses a modification of the algorithm of Lin and Vitter [19] for $k$-median. We remark that *any approximation algorithm* for $k$-median with $\ell_p$ objective can be used in place of the linear program (as we did in our experiments, for $p = \infty$, in which a greedy farthest point traversal can be used).

Consider a solution $(x, y)$ to the following linear programming (LP) relaxation:

$$\min \sum_u \sum_v d(u, v)^p x_{uv} \qquad \text{subject to}$$

$$\sum_v x_{uv} = 1 \qquad \text{for all } u \qquad \text{(every } u \text{ assigned to a center)}$$

$$x_{uv} \leq y_v \qquad \text{for all } u, v \qquad \text{(assigned only to center)}$$

$$\sum_u y_u \leq k \qquad \qquad \text{(at most } k \text{ centers)}$$

$$0 \leq x_{uv}, y_u \leq 1 \qquad \text{for all } u, v.$$

In the above algorithms, we can always treat $p \leq \log n$, and in particular the case $p = \infty$, as $p = \log n$. This introduces only negligible error in our computations but make them tractable. More specifically, when working with $p = \log n$, the power operators do not increase the size of the input by more than a factor $\log n$.

**Rounding** We perform a simple randomized rounding with weights scaled up by $O(\log n)$: round each $y_u$ to 1 with a probability equal to $\min\{1, y_u(4\log n)/\epsilon\}$. Let us denote this probability by $y'_u$, and the set of "centers" thus obtained, by $S$. We prove the following (proof in the supplement)

**Lemma 4.** *With probability $(1-1/n)$, the set $S$ of selected centers satisfies the following properties.*

1. *Each vertex has a relatively close selected center. In particular, for every $u \in V$, there is a center opened at distance at most $\left[(1+\epsilon)\sum_v d(u,v)^p x_{uv}\right]^{1/p}$.*

2. *Not too many centers are selected; i.e., $|S| < \frac{8k\log n}{\epsilon}$.*

*Mapping and multiplicity.* Once we have a set $S$ of centers, we map every $v \in V$ the center closest to it, i.e., $f(v) = \arg\min_{s \in S} d(v,s)$. If $m_s$ points in $V$ are mapped to some $s \in S$, we set its multiplicty to $m_s$. This defines a bijection from $V$ to the resulting multiset.

**Composability of the coresets.**

We now come to the crucial step, the proof of composability for the mapping coresets constructed earlier, i.e., the 'furthermore' part of Theorem 1.

To show this, we consider any vertex sets $V_1, V_2, \ldots, V_m$, and mapping coresets $S_1, S_2, \ldots, S_m$ obtained by the rounding algorithm above. We have to prove that the total moving cost is at most $(1+\epsilon)2^p OPT_{\mathcal{P}}$, where the optimum value is for the instance $\cup_i V_i$. We denote by $LP(V_i)$ the optimum value of the linear program above, when the set of points involved is $V_i$. Finally, we write $\mu_v := d(v, f_v)^p$, and $\mu_{\text{total}} := \sum_{v \in V} \mu_v$. We now have:

**Lemma 5.** *Let $LP_i$ denote the objective value of the optimum solution to $LP(V_i)$, i.e., the LP relaxation written earlier when only vertices in $V_i$ are considered. Then we have*

$$\mu_{total} \leq (1+\epsilon)\sum_i LP_i.$$

The proof follows directly from Lemma 4 and the definition of $f$. The next lemma is crucial: it shows that $LP(V)$ cannot be too small. The proof is deferred to the supplement.

**Lemma 6.** *In the notation above, we have $\sum_i LP_i \leq 2^p \cdot LP(V)$.*

The two lemmas imply that the total mapping cost is at most $(1+\epsilon)2^p OPT_{\mathcal{P}}$, because $LP(V)$ is clearly $\leq OPT_{\mathcal{P}}$. This completes the proof of Theorem 1.

# 5 Space efficient algorithms on a single machine

Our framework ultimately reduces distributed computation to a sequential computation on a compressed instance. For this, we need to adapt the known algorithms on balanced $k$-clustering, in order to handle compressed instances. We now give a high level overview and defer the details to the supplementary material.

For balanced $k$-center, we modify the linear programming (LP) based algorithm of [16], and its analysis to deal with compressed instances. This involves the following trick: if we have a compressed instance with $N$ points, since there are only $k$ centers to open, at most $k$ "copies" of each point are candidate centers. We believe this trick can be applied more generally to LP based algorithms.

For balanced $k$-clustering with other $\ell_p$ objectives (even $p = 1$), it is not known how to obtain constant factor approximation algorithms (even without the space efficient restriction). Thus we consider bicriteria approximations, in which we respect the cluster size constraints, but have up to $2k$ clusters. This can be done for all $\ell_p$ objectives as follows: first solve the problem approximately *without* enforcing the balanced constraint, then post-process the clusters obtained. If a cluster contains $n_i$ points for $n_i > L$, then subdivide the cluster into $\lceil n_i/L \rceil$ many clusters. The division should be done carefully (see supplement).

The post-processing step only involves the *counts* of the vertices in different clusters, and hence can be done in a space efficient manner. Thus the crucial part is to find the 'unconstrained' $k$-clustering in a space efficient way. For this, the typical algorithms are either based on local search (e.g., due

| Graph | Relative size of sequential instance | Relative increase in radius |
|---|---|---|
| US | 0.33% | +52% |
| World | 0.1% | +58% |

Table 2: Quality degradation due to the two-round approach.

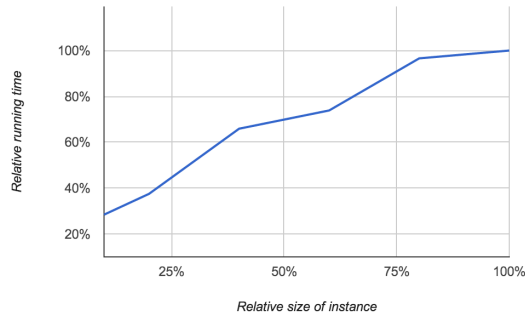

Figure 2: Scalability of parallel implementation.

to [13]), or based on rounding linear programs. The former can easily be seen to be space efficient (we only need to keep track of the number of centers picked at each location). The latter can be made space efficient using the same trick we use for $k$-center.

# 6  Empirical study

In order to gauge its practicality, we implement our algorithm. We are interested in measuring its scalability in addition to the effect of having several rounds on the quality of the solution.

In particular, we compare the quality of the solution (i.e., the maximum radius from the $k$-center objective) produced by the parallel implementation to that of the sequential one-machine implementation of the farthest seed heuristic. In some sense, our algorithm is a parallel implementation of this algorithm. However, the instance is too big for the sequential algorithm to be feasible. As a result, we run the sequential algorithm on a small sample of the instance, hence a potentially easier instance.

Our experiments deal with two instances to test this effect: the larger instance is the world graph with hundreds of millions of nodes, and the smaller one is the graph of US road networks with tens of millions of nodes. Each node has the coordinate locations, which we use to compute great-circle distances—the closest distance between two points on the surface of the earth. We always look for 1000 clusters, and run our parallel algorithms on a few hundred machines.

Table 2 shows that the quality of the solution does not degrade substantially if we use the two-round algorithm, more suited to parallel implementation. The last column shows the increase in the maximum radius of clusters due to computing the $k$-centers in two rounds as described in the paper. Note that the radius increase numbers quoted in the table are upper bounds since the sequential algorithm could only be run on a simpler instance. In reality, the quality reduction may be even less. In case of the US Graph, the sequential algorithm was run on a random $\frac{1}{300}$ subset of the actual graph, whereas a random $\frac{1}{1000}$ subset was used for the World Graph.

We next investigate how the running time of our algorithm scales with the size of the instance. We focus on the bigger instance (World Graph) and once again take its random samples of different sizes (10% up to 100%). This yields to varying instance sizes, but does not change the structure of the problem significantly, and is perfect for measuring scalability. Figure 2 shows the increase in running time is sublinear. In particular, a ten-fold increase in instance size only leads to a factor 3.6 increase in running time.

## Footnotes

[1]A main difference between the balanced graph partitioning problems and balanced clustering problems considered here is that in the graph partitioning problems a main objective function is to minimize the cut function.

[2]Here and elsewhere below, $\tilde{O}(\cdot)$ is used to hide a logarithmic factor.

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
