[Reviews · NeurIPS 2014]

Submitted by Assigned_Reviewer_6

This paper proposes coreset approach for balanced clustering. The paper is not clearly written and lacks intuition and motivation. Although it refers to clustering, what is exactly the clustering objective function? The problem defined in Sec 4 is slightly modified k-median problem, what does it has to do with cluster balance? The experiments part is next to none, did not compare with standard approach and show the class imbalance problem. This is very poorly written paper.
Summary: This paper is poorly written, no clear connection to cluster balance although it claim so, and the experiment part is extremely limited.

Submitted by Assigned_Reviewer_21

The paper describes an approach to distributed (balanced) clustering using the notion of a coreset, a concept introduced and studied in theoretical computer science in the areas of approximation algorithms and computational geometry. Starting from a recent work of Indyk et al. (PODS’14) the authors introduce the idea of “mapping coresets” which is basically a coreset together with a mapping from the the original space to the coreset points. Basically the proposed clustering algorithm works as follows: first the input data set is partitioned into several chunks arbitrarily. Then, for each chunk, a coreset is computed together with a mapping of the original points in the chunk to those in the coreset. Next, clustering is performed over the union of all the coresets and the solution found is finally mapped back to the original data points. After a long list of definitions the authors provide some theoretical properties and show connections to MapReduce and streaming clustering algorithms.

The writing of the paper is quite dense and technical, and it is not easy to read. Also, the authors presuppose familiarity with coresets and its properties and with results from approximation algorithms. Indeed, almost all references come basically from the theoretical computer science area (STOC, FOCS, SODA, etc.).

All in all the contribution is quite incremental. The idea of using coresets in distributed clustering has been used before extensively, although the idea introduced here of mapping coresets and the application to balanced clustering looks novel. The connection with MapReduce and streaming clustering algorithm is interesting but I'm not sure this will appeal to the NIPS community.

A major weakness of the paper, however, is the experimental evaluation which aimed to show the scalability of the proposed algorithm. Indeed, the authors applied their implementation of the algorithm on just two (admittedly very large) graphs, so the overall conclusions are questionable.

Summary: All in all, I think this paper is not suitable for presentation at NIPS. It’s an incremental piece of theoretical work whose impact in the NIPS community is questionable. I think a more appropriate audience for this kind of papers is the approximation algorithm community

Submitted by Assigned_Reviewer_41

This paper considers distributed balanced clustering problem, where the size of each cluster has an upper bound, and the data is under distributed storage. In theory, if not consider distribution, the problem is also called ``capacitated clustering", which has been well studied before, such as [10][16]. Here, the authors mainly address two issues, one is how to compress the large scale input into a small set under distributed storage, and the other is how to fit the computation into Mapreduce framework. For the first one, they present a new type of coreset, which is called ``composable coreset". The advantage is that composable coreset can be constructed under distributed storage, and the objective cost is increase by a factor of 2^p. For the latter issue, they modify the computation model for mapreduce introduced by [15]. In [15], each machine has N^{1-\gamma} memory, but for the algorithm presented in this paper, it needs quadratic size of memory. Therefore, they show a two-level framework, where each level involves partition and coreset construction steps. After addressing the mapping coreset framework, they introduce the algorithms in section 4 and 5, which are based on the previous methods from [19][16]. They provide a theoretical guarantee for their method, which is a 32\alphaI approximation. In the experiment, the authors test their method on two datasets, world graph and US road network, and evaluate the results based on both quality and scalability.
Summary: In general, this paper presents a novel approach for a clustering problem under distributed storage, which is quite common in big data. Both of theoretical analysis and experiment are solid, and I think it is a good paper for publishing in NIPS.
Author Feedback
Author rebuttal: We thank the reviewers for their thoughtful comments and suggestions. We identified the main concerns and we address those first. Then we will address all other comments. Following your comments we believe we can substantially improve the exposition to make it more accessible to NIPS audience and to clarify our contributions to NIPS community.

1. Incremental
Core-sets are a well-established tool to solve clustering problems, but they have never been applied to solve balanced (i.e., capacitated) problems. Indeed, capacitated clustering has long proved challenging from an approximation standpoint, and most known methods in the algorithms literature are based on linear programming, and are difficult to parallelize. The techniques introduced in section 4 are novel and could potentially be used in analyzing other core-set based algorithms.

Finally, to the best of our knowledge, our paper gives the first algorithms for balanced k-center, k-median and k-means problems that have provable guarantees and that are scalable. Even for the problem without capacity constraints, our framework gives a new general tool to solve these clustering problems in parallel.

2. Motivations
k-center, k-median and k-means are classic clustering problems and have been extensively used in a variety of applications. In many of the natural applications, there is an implicit “capacity” constraint, e.g., for the purposes of load balancing, or simply to obtain a well-balanced partitioning of the input.

3. Relevance to NIPS audience
Clustering (with k-center, k-median and k-means objectives) is a fundamental subroutine used in a variety of applications in learning and data mining. We provide below a short list of NIPS papers published after 2009 that analyze those problems:
(a) Distributed k-means and k-median clustering on general communication topologies by M. Balcan, S. Ehrlich, Y. Liang
(b) Moment-based Uniform Deviation Bounds for k-means and Friends by M. Telgarsky, S. Dasgupta
(c) Fast and Accurate k-means For Large Datasets by M. Shindler, A. Wong, A. Meyerson
(d) Random Projections for $k$-means Clustering by C. Boutsidis, A. Zouzias, P. Drineas
(e) Streaming k-means approximation by N. Ailon, R. Jaiswal, C. Monteleoni
(f) Unsupervised Feature Selection for the $k$-means clustering problem by C. Boutsidis, M. Mahoney, P. Drineas

We also note that 3 out of the 6 listed results propose streaming/distributed/largescale algorithm for those problems. Our paper introduces a simple yet general technique that can be used to obtain interesting bounds in all those settings and additionally extends to the balanced version of the problem. Our framework comes with theoretical approximation guarantees, and also scales well in practice. Thus we believe it is of more relevance to a NIPS audience than a theory/algorithms one. In the final version, we will include more references to work in learning and data mining that have (distributed) clustering as a key component. We will also provide more intuitive descriptions that precede formal proofs.

4. Experiments
Our goal in the experimental section was two-fold: first, we wanted to demonstrate that that our techniques are not only interesting in theory but they are also practical (and scale well). Second, we observe that in practice our algorithms could obtain a much better approximation factor than our theoretical (worst-case) results can establish.

5. Problem Definition
We note that we formally define the problem in Section 2 (Preliminaries), and present rigorous mathematical descriptions of the objective function and the constraints.

6. Minor comments:
"Although it refers to clustering, what is exactly the clustering objective function?" Problem definition is clearly defined in Section 2, an entire section is dedicated to it; see, for instance, Equation (1).

"The problem defined in Sec 4 is slightly modified k-median problem, what does it has to do with cluster balance?" Section 4, as its title suggests, describes our core-set construction and not our problem statement. The latter is done in Section 2.

"No clear connection to cluster balance although it claim so" Definition 1 defines a balanced clustering problem and Sections 3, 4 and 5 describe how to solve this problem. One of the algorithmic tools used in the paper is based on a non-balanced formulation of the problem. Nevertheless the framework presented in the paper gives a solution for the balanced clustering problem described in Definition 1.